# PAK1-Dependent Regulation of Microtubule Organization and Spindle Migration Is Essential for the Metaphase I–Metaphase II Transition in Porcine Oocytes

**DOI:** 10.3390/biom14020237

**Published:** 2024-02-17

**Authors:** Lei Peng, Yijing He, Weihan Wang, Jianjun Dai, Qiao Li, Shiqiang Ju

**Affiliations:** 1MOE Joint International Research Laboratory of Animal Health and Food Safety, College of Veterinary Medicine, Nanjing Agricultural University, Nanjing 210095, China; pengleinjau@163.com (L.P.); 2022207021@stu.njau.edu.cn (Y.H.); wwang@njau.edu.cn (W.W.); 2Key Laboratory of Livestock and Poultry Resources (Pig) Evaluation and Utilization, Ministry of Agriculture and Rural Affairs, Institute of Animal Husbandry and Veterinary Science, Shanghai Academy of Agricultural Sciences, Shanghai 201106, China; blackman0520@126.com

**Keywords:** porcine oocytes, P21-activated kinase 1, metaphase I–metaphase II transition, spindle dynamics

## Abstract

P21-activated kinase 1 (PAK1) is a critical downstream target that mediates the effect of small Rho GTPase on the regulation of cytoskeletal kinetics, cell proliferation, and cell migration. PAK1 has been identified as a crucial regulator of spindle assembly during the first meiotic division; however, its roles during the metaphase I (MI) to metaphase II (MII) transition in oocytes remain unclear. In the present study, the potential function of PAK1 in regulating microtubule organization and spindle positioning during the MI–MII transition was addressed in porcine oocytes. The results showed that activated PAK1 was co-localized with α-tubulin, and its expression was increased from the MI to MII stage (*p* < 0.001). However, inhibiting PAK1 activity with an inhibitor targeting PAK1 activation-3 (IPA-3) at the MI stage decreased the first polar body (PB1) extrusion rate (*p* < 0.05), with most oocytes arrested at the anaphase-telophase (ATI) stage. IPA-3-treated oocytes displayed a decrease in actin distribution in the plasma membrane (*p* < 0.001) and an increase in the rate of defects in MII spindle reassembly with abnormal spindle positioning (*p* < 0.001). Nevertheless, these adverse effects of IPA-3 on oocytes were reversed when the disulfide bond between PAK1 and IPA-3 was reduced by dithiothreitol (DTT). Co-immunoprecipitation revealed that PAK1 could recruit activated Aurora A and transform acidic coiled-coil 3 (TACC3) to regulate spindle assembly and interact with LIM kinase 1 (LIMK1) to facilitate actin filament-mediated spindle migration. Together, PAK1 is essential for microtubule organization and spindle migration during the MI–MII transition in porcine oocytes, which is associated with the activity of p-Aurora A, p-TACC3 and p-LIMK1.

## 1. Introduction

P21-activated kinase 1 (PAK1) is involved in the regulation of processes associated with cytoskeletal remodeling that affect cell division, migration, and survival [1]. In cell division, PAK1 plays a role in the regulation of microtubule and actin filament dynamics [2,3]. During mitosis, the activation of PAK1 is mediated by PAK-interacting exchange factor beta (β-PIX) and G-protein coupled receptor-interacting protein 1 (GIT1), which can promote the activation of Aurora A and stabilize the centrosome dynamics in metaphase [4]. During the prophase of mitosis, the depletion of PAK1 by siRNA could lead to an increased intercentrosomal distance in Madin–Darby canine kidney strain II cells [5]. Moreover, it has been found that PAK1 regulates the aggregation of the actin filaments during meiosis [6]. In the process of actin dynamics, PAK1 is activated by cell division control protein 42 (Cdc42) GTPase, which then phosphorylates LIM kinase 1 (LIMK1) at threonine 508, resulting in a decreased depolymerization of F-actin [7]. PAK1 also drives cytoskeletal reorganization, causing a more rapid depolymerization of filamentous actin [8]. However, an inhibitor targeting PAK1 activation-3 (IPA-3) leads to the degeneration of actin filaments in prostate stromal cells [9].

In meiosis, metaphase I (MI) is followed by metaphase II (MII) without an intervening synthesis phase [10]. During the stages from MI to MII, the oocytes form bipolar spindles independent of the centrosome, exhibit an asymmetric division to segregate homologous chromosomes, and then arrest at the MII stage to wait for further fertilization [11]. Additionally, microtubules and actin filaments are involved in the organization and orientation of the spindles, which are mediated by Ran GTP and actin-related proteins to achieve an accurate segregation of the maternal genome [12,13]. Any errors in the MI–MII transition can cause chromosomal aneuploidy and embryonic development defects.

During the first meiotic division, the inhibition of PAK1 activity in mouse oocytes leads to extreme spindle disassembly and chromosome misalignment [14,15]. According to our previous study, the treatment of IPA-3 on the germinal vesicle (GV) stage in porcine oocytes causes a failure of bipolar spindle formation during meiosis I, affecting oocyte maturation [16]. Although there has been much information about the function of PAK1 in mitosis and spindle dynamics during meiosis I [17,18,19], little is known about its effects during the MI–MII transition. In this study, the activity of PAK1 was inhibited by the specific inhibitor IPA-3 in the MI oocytes to explore the mechanism for the roles of PAK1 during the transformation of MI–MII in porcine oocytes.

## 2. Materials and Methods

### 2.1. Antibodies and Chemicals

Anti-PAK1 pSer204 antibody (11748), anti-PAK1 antibody (Ab-212), anti-Aurora A pThr288 antibody (12301), anti-TACC3 pSer558 antibody (13495), and anti-LIMK1 pThr508 (11123) antibody were purchased from Signalway Antibody (College Park, MD, USA). Anti-Aurora A antibody was purchased from Abcam (Ab13824, Cambridge, UK). Anti-TACC3 (SN73-05) and Rabbit polyclonal anti-LIMK1 (HA500189) antibodies were purchased from Sigma-Aldrich (St. Louis, MO, USA). IPA-3 was purchased from Selleck Chemicals (S7093, Houston, TX, USA). All other chemicals and reagents were purchased from Sigma-Aldrich, unless explicitly specified or indicated otherwise.

### 2.2. Oocytes Collection and Culture

Porcine ovaries were acquired from prepubertal gilts at a local slaughterhouse and conveyed to the laboratory in sterile 0.9% (*w*/*v*) physiological saline within 2 h. The cumulus oocyte complexes (COCs) with integrated cumulus cells and homogeneous cytoplasm were selected from follicles with diameters ranging from 3 to 6 mm. These COCs were then transferred to pre-equilibrated TCM199 medium (Gibco BRL, Gaithersburg, MD, USA), containing 3.05 mM D-glucose, 0.1% (*w*/*v*) polyvinyl alcohol, 0.91 mM sodium pyruvate, 10 IU/mL PMSG and hCG, 0.57 mM cysteine, 10 ng/mL epidermal growth factor (EGF), 10% (*v*/*v*) porcine follicular fluid (pFF), 5.0 mg/mL streptomycin, and 7.5 mg/mL penicillin, and then the COCs were cultured at a temperature of 38.5℃ in an environment with 5% CO_2_ and saturated humidity, following a previously reported study [20]. The majority of oocytes developed to GV, MI, anaphase-telophase (ATI), and MII after 0, 28, 36, and 44 h of culture were collected for further investigation [21,22].

### 2.3. Immunofluorescence Staining

Denuded oocytes were immobilized in 4% paraformaldehyde for 30 min and then permeabilized in 1% Triton X-100 for 8 h. Following an hour of incubation in 1% bovine serum albumin (BSA), the oocytes were subjected to an overnight incubation at 4℃ with primary antibodies. And then the oocytes were incubated with TRITC-labeled goat anti-rabbit IgG. Next, oocytes were stained with Phalloidin-Tetramethylrhodamine B isothiocyanate for 40 min or anti-α-tubulin antibody (1:200) for 2 h at 37 °C. Then, the oocytes underwent staining with Hoechst 33342 (10 μg/mL) for a duration of 10 min for DNA counterstaining. Finally, the oocytes were detected using laser scanning microscope (Zeiss LSM700 META, Oberkochen, Germany) and analyzed with Image J 1.8.0 (National Institutes of Health, Bethesda, MD, USA).

### 2.4. Immunoblotting

The oocytes, with an approximate count of 100 per group, were lysed using Laemmli buffer and heated at 95 °C for 10 min. Cell lysates were separated using 10% SDS-PAGE and proteins were transferred into polyvinylidine fluoride (PVDF) membranes (Millipore, Billerica, MA, USA). The membrane was blocked with 5% nonfat-dried milk and then incubated with the primary antibodies and appropriate secondary antibodies. The membrane was subjected to visualization using a chemiluminescence reagent and analyzed with Image J 1.8.0 software (National Institutes of Health, USA).

### 2.5. Co-Immunoprecipitation (Co-IP)

The Co-IP experiment was performed using the IP/Co-IP kit (Absin, Shanghai, China) in accordance with the instructions provided by the manufacturer. A total of 700 oocytes in each group were lysed using lysis buffer (500 µL). Subsequently, the cell lysate was subjected to incubation with a 3 µL volume of anti-PAK1 antibody at a temperature of 4 °C for the duration of the overnight period. Following this, protein A and G beads (5 µL) were used to deposit the complex. Subsequently, the proteins were degenerated using SDS loading buffer and then subjected to immunoblotting.

### 2.6. Experimental Design

#### 2.6.1. Effects of IPA-3 Treatment during the MI–MII Transition

A total of 40 oocytes in each group were placed into TCM-199 medium for 28 h. Then, final concentrations of 0, 10, 20, and 30 μM IPA-3 were introduced into the subsequent culture. After 44 h of culture, the first polar body (PB1) extrusion and the meiotic progression of the oocytes were evaluated. The expression and localization of p-PAK1 in the oocytes after 0, 28, 36, and 44 h of culture were also detected. Both the 0 μM and 20 μM IPA-3 groups were subjected to immunofluorescence labeling in order to evaluate the subcellular structure of spindles (*n* = 40) and actin (*n* = 30) following a 44 h culture period.

#### 2.6.2. Effects of Dithiothreitol (DTT) on IPA-3 Inhibition

On the basis of the effect of PAK1 inhibition on meiotic progression, most oocytes were arrested at the ATI stage with IPA-3 treatment after 36 h of culture. DTT is regarded as a reductive agent that acts upon the disulfide link existing between PAK1 and IPA-3 [23]. Therefore, the oocytes in the IPA-3 + DTT group were treated with IPA-3 (20 μM) for 36 h and then transferred to TCM 199 medium with different concentrations of DTT (0.25, 0.5, and 1 mM) for 13 min [23]. Finally, the oocytes were transferred into normal TCM199 medium as IPA-3 + DTT group. Following 44 hours of culture, the maturation of oocytes was assessed for each group (consisting of 40 oocytes each). According to the results obtained, 0.5 mM DTT was used for further immunoblotting and immunofluorescence staining experiments. Ultimately, Co-IP was used to identify the interaction between Aurora A, TACC3, and LIMK1.

### 2.7. Statistical Analysis

IBM SPSS (Statistics Production for Service Solution, Version 22.0) was used to analyze experimental data. Each experiment was conducted in three replicates. One-way ANOVA and Duncan’s multiple comparisons tests were used to test the difference between groups. The results are shown as means ± standard error and a *p* value below 0.05 was considered significant.

## 3. Results

### 3.1. Expression and Localization of p-PAK1

As shown in Figure 1A,B, the expression of p-PAK1 was detected. A lower p-PAK1 level was found at 28 h (*p* < 0.001), but a higher level was observed at 44 h (*p* < 0.05) when compared with that at 0 h. The α-tubulin and p-PAK1 were observed to disseminate around the germinal vesicle at 0 h of culture (Figure 1C). From 28 h to 36 h of culture, homologous chromosomes were pulled by the barrel-like bipolar spindles to two distinct poles. Simultaneously, the presence of p-PAK1 was seen in close proximity to α-tubulin, showing enrichment in the spindle area. After being cultivated for a duration of 44 h, oocytes underwent extrusion of PB1. Additionally, α-tubulin molecules were organized into the MI spindle, which was subsequently dispersed at the location of the PB1. The distribution of p-PAK1 was seen in close proximity to the spindle inside the cytoplasm of the oocyte as well as the polar body. These results indicated that p-PAK1 co-localized with the meiotic spindle during the MI–MII meiotic progression in porcine oocytes.

### 3.2. IPA-3 Treatment Caused Oocytes Maturation Failure and Cell Cycle Arrest during the MI–MII Transition

To investigate the potential functions of PAK1 during the transition from MI to MII, the oocytes were treated with IPA-3 at 28 h of culture. As shown in Figure 2B,C, the rate of PB1 extrusion was higher in the IPA-3-treated group compared to the control group (Control vs. 20 μM IPA-3 vs. 30 μM IPA-3 groups: 80.00 ± 2.9% vs. 61.67 ± 1.7% vs. 42.50 ± 1.4%, *p* < 0.01). In addition, Figure 2D,E present that IPA-3 perturbed meiotic progression; most oocytes in the IPA-3-treated group were maturation failures and arrested at the ATI stage (Control vs. 20 μM IPA-3 vs. 30 μM IPA-3: 0.83 ± 0.8% vs. 13.33 ± 0.8% vs. 20.83 ± 0.8%, *p* < 0.001). According to the results mentioned above, a concentration of 20 μM IPA-3 was used for further exploration. As shown in Figure 2F,G, the expression of p-PAK1 was reduced following treatment with 20 μM IPA-3 compared with the control group (*p* < 0.05).

### 3.3. IPA-3 Treatment Perturbed Cytoskeletal Dynamic of Porcine Oocytes during the MI–MII Stage

In order to analyze the possible underlying explanation for the inability of IPA-3-treated oocytes to advance to the MII stage, an investigation was conducted to analyze the spindle morphology of oocytes. After 44 h of culture, most of the oocytes in the control group displayed normal bipolar spindle and a polar body. However, Figure 3A,B showed a markedly higher percentage of defected spindles in the IPA-3-treated group, such as multipolar spindle and failed separation, than that of the control group (37.50 ± 1.4% vs. 16.67 ± 1.7%, *p* < 0.001), with those disorganized spindles displayed at a position distant from the cortex. Consequently, oocytes subjected to PAK1 inhibition experienced arrest at the ATI stage, rendering them unable to advance to the MII stage. This arrest was attributed to the inability to reassemble the MII spindle and the occurrence of aberrant migration during the transition from MI to MII. The spindle migration is mediated by the distribution of actin filaments. Next, we explored the effect of IPA-3 on actin distribution in porcine oocytes during the MI–MII transition. As shown in Figure 3C,D, the actin distribution was disturbed both at the membrane and cytoplasm of oocytes after IPA-3 treatment, mainly manifested by the disrupted structure of the actin cap, and mislocalized actin filaments in the cytoplasm and the actin filament signals at the plasma membrane decreased visibly in IPA-3-treated oocytes (*p* < 0.001). These results indicated that IPA-3 treatment caused an abnormal spindle migration due to its effect on actin dynamics.

### 3.4. DTT Recovered the Oocytes’ Maturation and Normal Cytoskeletal Dynamics

According to the data depicted in Figure 4B,C, a greater proportion of oocytes subjected to PAK1 inhibition exhibited a failure to extrude PB1 and progressed to the MII stage. Conversely, the majority of oocytes effectively extruded the PB1 following the IPA-3 and DTT cotreatment, therefore reaching a similar level as the control group. In comparison to the IPA-3 group (65.00 ± 1.4%, *p* < 0.01), the PB1 extrusion rate of the oocytes inhibited with PAK1 exhibited a substantial rise to 80.83 ± 0.8% upon treatment with 0.5 mM DTT (83.33 ± 3.0%, *p* > 0.05). Furthermore, following the administration of 0.5 mM DTT, there was a reduction in the percentage of cell cycle arrested in the IPA-3 and DTT group compared to the IPA-3-treated groups (14.17 ± 1.7% vs. 1.67 ± 1.7%, *p* < 0.001, Figure 4D,E).

Based on these results, 0.5 mM DTT was used for subsequent studies to repair the inhibition of IPA-3 on PAK1 activity. The expression of p-PAK1 in the IPA-3 + DTT group was increased (*p* < 0.01, Figure 4F,G) compared with that in the IPA-3 treatment group after a total of 44 h. In the IPA-3 + DTT group, the percentage of abnormal spindles had also markedly decreased compared to the IPA-3 treatment group (37.50 ± 3.8% vs. 15.83 ± 2.2%, *p* < 0.01, Figure 5A,B), and the spindle in the IPA-3 + DTT group exhibited a peripheral localization. Moreover, the distribution of actin filaments was restored after 0.5 mM DTT treatment, in which the actin cap was formed and action filaments were normally distributed at the membrane of the oocytes, and the relative fluorescence intensity of membrane actin in the DTT group was higher than that of the IPA-3 group (Figure 5C,D, *p* < 0.001).

### 3.5. IPA-3 Treatment Decreased the Activity of Aurora A, TACC3, and LIMK during the MI–MII Transition

To further determine the possible mechanism of PAK1 in porcine during the MI–MII transition, the interactions between PAK1 and its substrates, Aurora A, TACC3, and LIMK, were examined. Firstly, the results of Co-IP exhibited that PAK1 was associated directly with Aurora A, TACC3, and LIMK1 during the MI–MII transition (Figure 6A). Meanwhile, DTT could restore the inhibition of the expression of p-Aurora A, p-TACC3, and p-LIMK1 in the IPA-3 group (Figure 6B–E, *p* < 0.05). Then, the subcellular localization and fluorescence intensity of p-Aurora A, p-TACC3, and p-LIMK1 were also analyzed. The data showed that IPA-3 treatment resulted in a decreased fluorescence intensity of p-Aurora A, p-TACC3, and p-LIMK1 in the cytoplasm of oocytes (Figure 6F–K, *p* < 0.001). However, the IPA-3 inhibition could be restored by DTT administration (Figure 6F–K, *p* < 0.001).

## 4. Discussion

Although there is much information regarding the role of PAK1 in mitosis and spindle dynamics during meiosis I, the exact mechanism of PAK1 regulation in the MI–MII transition of mammalian oocytes has not yet been demonstrated. In this study, the possible roles of PAK1 in porcine oocytes during the MI–MII transition were identified. The data indicated that PAK1 contributed to the successful meiotic maturation of porcine oocytes by regulating the proper MII spindle reassembly and the accurate orientation of the spindle. This underlying regulation might be associated with PAK1 positive effects on the microtubule-related proteins Aurora A and TACC3, as well as the actin-related protein LIMK1.

The PAK family consists of six members, PAK1 to PAK6 [24]. PAK1, PAK2, and PAK4 have been reported to exist in mammalian oocytes [15,25]. In the current study, we identified that activated PAK1 exhibited a dynamic distribution and expression during meiosis in porcine oocytes. Activated PAK1 was observed to accumulate around the spindle region at the MI and ATI stages, which is consistent with the subcellular localization pattern of microtubule-associated proteins in meiosis, such as NuMA [26] and HURP [27], while activated PAK1 was enriched at the α-tubulin of the first polar body during meiosis II, which is similar to the localization of the actin-related protein 2/3 (Arp2/3) complex in porcine oocytes [28]. Therefore, PAK1 may be involved in the regulation of spindle formation and actin filaments during the MI–MII transition in porcine oocytes.

The treatment of mouse oocytes with IPA-3, a specific inhibitor of PAK1, blocked the extrusion of the first polar body in mouse oocytes with a disorganized MI cytoskeleton [15]. Moreover, our previous research has reported that when PAK1 activity was inhibited at the GV stage, the porcine oocytes failed maturation and were arrested at the GVBD stage with the abnormal spindle structure [16]. Both findings demonstrated a crucial role of PAK1 in meiotic maturation before the MI stage. To examine the potential role of PAK1 during the MI–MII transition, oocytes were treated with IPA-3 from the MI stage. Our data showed that IPA-3 treatment significantly inhibited PAK1 activity, which caused the meiotic progression to be arrested at the ATI stage, Most of the PAK1-inhibited oocytes exhibited abnormal spindle migration and MII spindle reassembly. Based on these results, PAK1 is involved in the assembly of the spindle during meiosis. Similarly, a previous report on mitosis suggested that targeting PAK1 using IPA-3 could significantly inhibit the murine metastatic PCa cell proliferation and motility in vitro [29]. Actin filaments have been involved in the asymmetric division in mammalian oocytes through the regulation of vesicle transport, spindle migration, and anchorage [30]. During meiosis, chromosome movement to the cortex of oocytes is perturbed when the actin polymerization is inhibited by cytochalasin B [31]. It was also reported that PAK1 inhibition by FRAX1036 caused the abnormal structure and spatial position of F-actin bundles [8]. In the current study, IPA-3-treament disrupted the distribution of actin filaments, which led to spindle positioning near the center of the oocytes. Hence, we hypothesize that PAK1 has a potential role in the regulation of MII spindle reassembly and spindle migration facilitated by actin filaments during the MI–MII transition in porcine oocytes.

Previous studies showed that DTT can reduce the disulfide, reversing the inhibition of PAK1 activity by IPA-3 [23], and that the binding between PAK1 and IPA-3 was markedly weakened after treatment with 0.1 to 20 mM DTT for 13 min in vitro [32]. Consistently with these results, we found that the decreased expression of activated PAK1 in IPA-3-treated oocytes was restored in the 0.25 mM DTT group. In addition, the reassembly of the MII spindle in the PAK1-inhibited oocytes was returned to normal, and most of the oocytes in the IPA-3 and DTT cotreatment group exhibited an ordered polymerization of actin filaments at the cortex, restoring the migration of the spindle in a normal manner. As a result, when restoring PAK1 function, failed oocyte maturation and arrested cell cycle progress in the PAK1-inhibited oocytes were considerably recovered, while the percentage of oocytes arrested at the ATI stage was drastically decreased. Both findings indicate that PAK1 plays a pivotal role in the reorganization of the MII spindle and spindle location during the MI–MII transition in porcine oocytes.

In mitosis, PAK1 is shown to interact with Aurora A to regulate the organization of the bipolar spindle [33]. PAK1 phosphorylated Aurora A on Thr-288, and its inhibition of PAK1 by ectopic expression of kinase inhibitory domain could result in the delayed activation of Aurora A during mitosis [33,34]. Aurora A has been identified as an indispensable kinase for mitotic entry and centrosome maturation [35]. During interphase, Aurora A participates the assembly of centrosome, then localizes to the spindle during the mitosis, and finally degrades at the mitosis anaphase [36]. The specific inhibition of Aurora A by GSK6000063A in interphase results in microtubule network disorganization and bundling, which affects the dynamics and nucleation of microtubules [37]. Likewise, our Co-IP test revealed a direct interaction between PAK1 and Aurora A, and the inhibition of PAK1 by IPA-3 resulted in the suppression of the activity of Aurora A. TACC3, a member of the TACC family, plays an essential role on centrosome-mediated microtubule nucleation in mitosis, which requires the activation of Aurora A [38]. The inhibition of Aurora A kinase by MLN8237 has been identified to disrupt the function of TACC3, resulting in the destabilization of kinetochore fibers and the loss of inter-microtubule bridges [39]. The activated Aurora A can promote the spindle recruitment of TACC3 and triggers the formation of a complex with ch-TOG-clathrin that crosslinks and stabilizes kinetochore microtubules [40]. Surprisingly, we found that TACC3 presents as a potential downstream protein of PAK1 and interacts directly with it, while the expression and activation of TACC3 is suppressed with the inhibition of PAK1. In addition, our previous study found that TACC3 is present at the same subcellular localization as PAK1 [16]. PAK1 also appears to regulate the polymerization and density of the F-actin cytoskeleton to facilitate migration [41]. The regulation of PAK1 activity in cytoskeletal dynamics is related to its interaction with its substrate, LIMK1 [7]. A prior study found that PAK1 knockdown in neurons caused a decrease in the expression of activated LIMK1, interfering with normal F-actin remodeling [42]. The overexpression of PAK1 could translocate LIMK1 to the membrane and enhance F-actin polymerization [43]. Our study also found a direct interaction between PAK1 and LIMK1, and the inhibition of PAK1 by IPA-3 resulted in the suppression of the activity of LIMK1 in meiosis. Therefore, these findings indicate that PAK1 affects spindle formation through its interaction with Aurora A and TACC3, and interferes with actin filament-mediated spindle migration by regulating LIMK1 during the MI–MII transition in porcine oocytes.

## 5. Conclusions

The findings of this study indicate that the presence of PAK1 is crucial for the accurate assembly and movement of the meiotic spindle during the transition from metaphase I to metaphase II in pig oocytes. In addition, the regulation of PAK1 during the MI–MII transition is associated with its interaction with Aurora A, TACC3, and LIMK1. Our study demonstrates the regulatory function of PAK1 in oocyte cytoskeleton dynamics, which enriches our knowledge about of the mechanism of meiosis during porcine oocyte maturation.

## Figures and Tables

**Figure 1 biomolecules-14-00237-f001:**
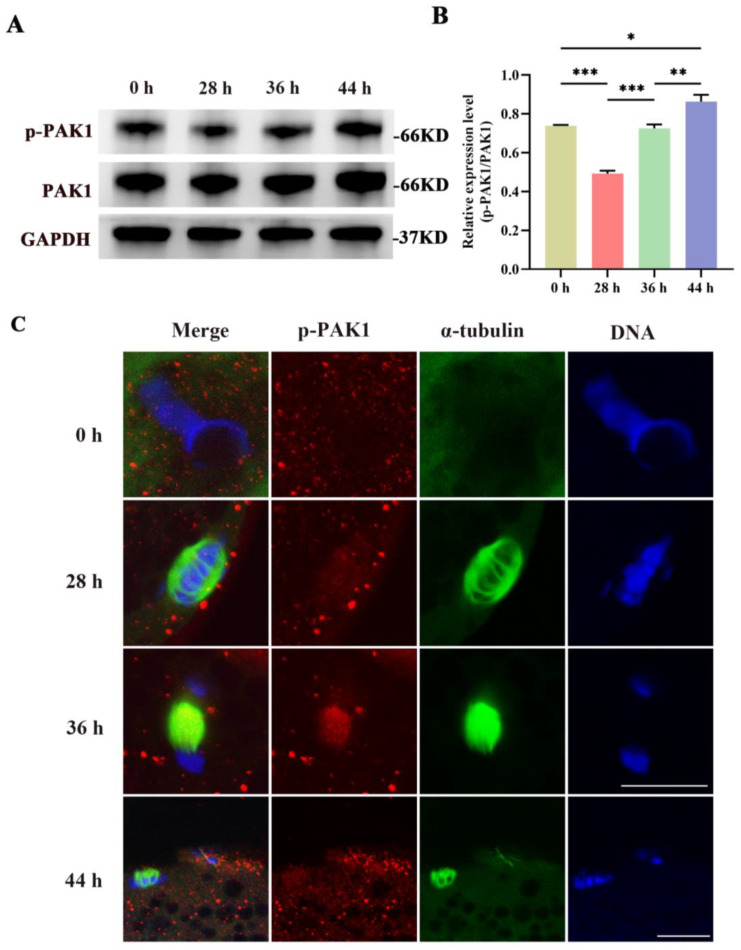
(**A**) Representative images of p-PAK1 and total PAK1 expression in porcine oocytes after 0, 28, 36, and 44 h of culture using immunoblotting. (**B**) The analysis of p-PAK1/PAK1 expression in oocytes. (**C**) Representative images of the subcellular localization of p-PAK1 in oocytes after 0, 28, 36, and 44 h of culture. Red, p-PAK1; green, α-tubulin; blue, chromosome. Scale bar = 20 μm. PAK1, P21-activated kinase 1; p-PAK1, phosphorylated P21-activated kinase 1. *, *p* < 0.05; **, *p* < 0.01; ***, *p* < 0.001. Original blot images can be found in Appendix A.

**Figure 2 biomolecules-14-00237-f002:**
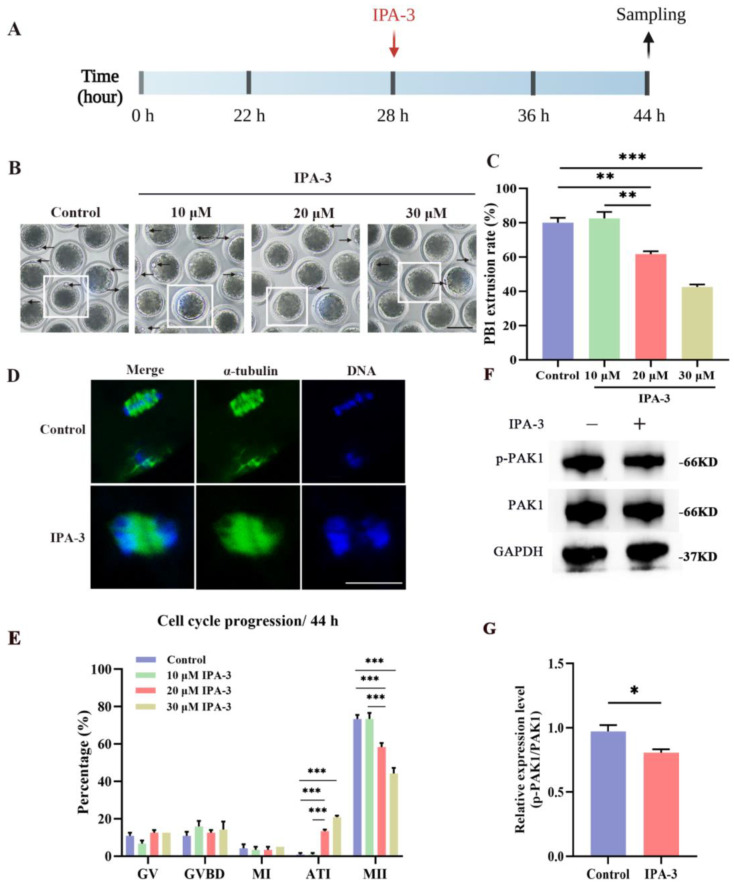
(**A**) The schematic diagram for the experimental design of the IPA-3 treatment. (**B**) Representative images of the PB1 extrusion with 10, 20, and 30 μM IPA-3 treatment. Arrow, the PB1. Scale bar = 100 μm. (**C**) The rates of PB1 extrusion in different groups. (**D**) Representative images of the cytoskeleton with IPA-3 treatment. Green, α-tubulin; blue, chromosome. Scale bar = 20 μm. (**E**) The analysis of cell cycle progression in different groups. *n* = 120. (**F**) Representative images of p-PAK1 expression after 28 h of culture. (**G**) The analysis of p-PAK1/PAK1 expression in oocytes. IPA-3, inhibitor targeting PAK1 activation-3; PB1, the first polar body; GV, germinal vesicle; GVBD, germinal vesicle breakdown; MI, metaphase I; ATI, anaphase-telophase I; MII, metaphase II. The letter “*n*” means the total number of oocytes in each group. *, *p* < 0.05; **, *p* < 0.01; ***, *p* < 0.001. Original blot images can be found in Appendix A.

**Figure 3 biomolecules-14-00237-f003:**
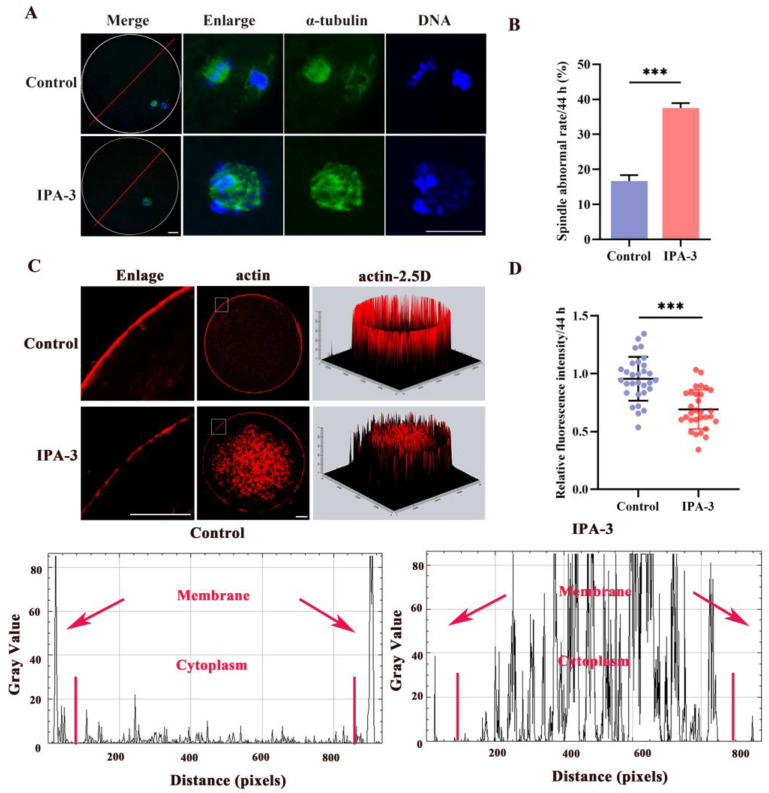
(**A**) Representative images of spindle morphology (green) and chromosome alignment (blue). Scale bar = 20 μm. (**B**) Percentages of abnormal spindles in different groups. *n* = 120. (**C**) Representative image of actin filaments in control and IPA-3-treated oocytes. Scale bar = 20 μm. (**D**) Quantification of the immunofluorescence intensity level of actin filaments in the control and IPA-3-treated oocytes. *n* = 30. The letter “*n*” means the total number of oocytes in each group. ***, *p* < 0.001.

**Figure 4 biomolecules-14-00237-f004:**
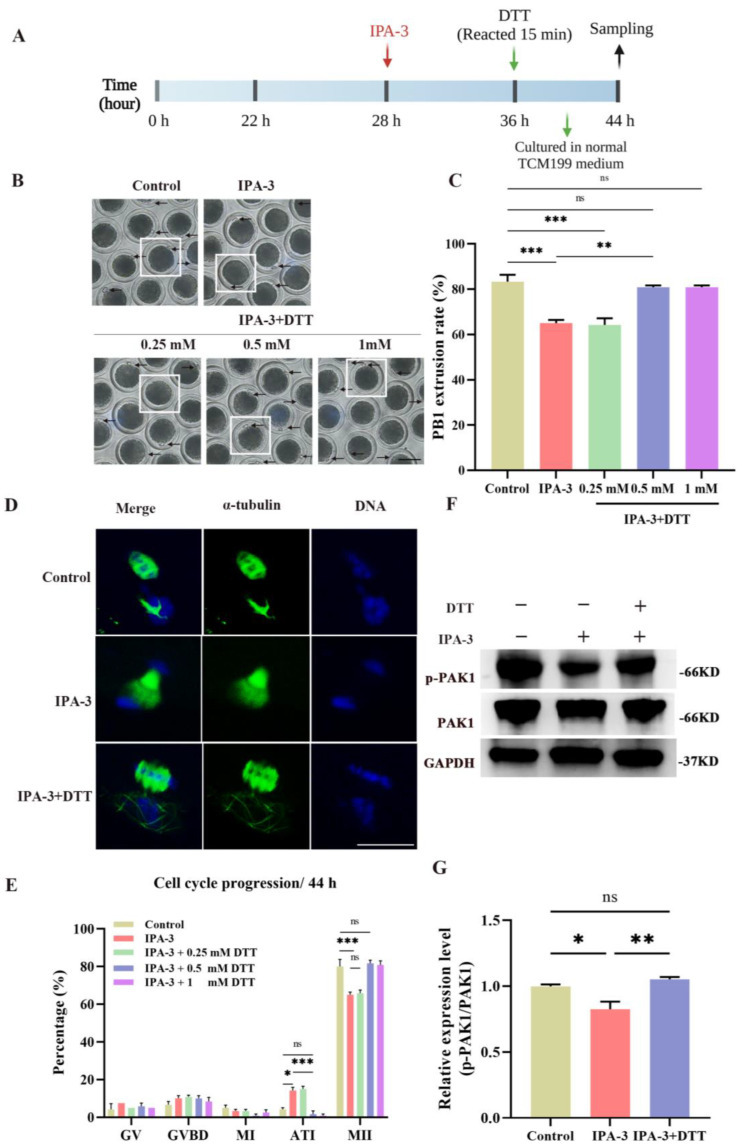
(**A**) The schematic diagram for the experimental design of DTT rescue. (**B**) Representative images of the PB1 extrusion with 20 μM IPA-3 and different doses of DTT treatment. Arrow, the PB1. Scale bar = 100 μm. (**C**) The PB1 extrusion rate in different groups. *n* = 120. (**D**) Representative images of the cytoskeleton with 20 μM IPA-3 and different doses of DTT treatment. Green, α-tubulin; blue, chromosome. Scale bar = 20 μm. (**E**) The analysis of cell cycle progression in different groups. *n* = 120. (**F**) Representative images of p-PAK1 expression after 44 h of culture using immunoblotting. (**G**) The analysis of p-PAK1/PAK1 expression in oocytes. DTT, dithiothreitol. The letter “*n*” means the total number of oocytes in each group. ns, *p* > 0.05; *, *p* < 0.05; **, *p* < 0.01; ***, *p* < 0.001. Original blot images can be found in Appendix A.

**Figure 5 biomolecules-14-00237-f005:**
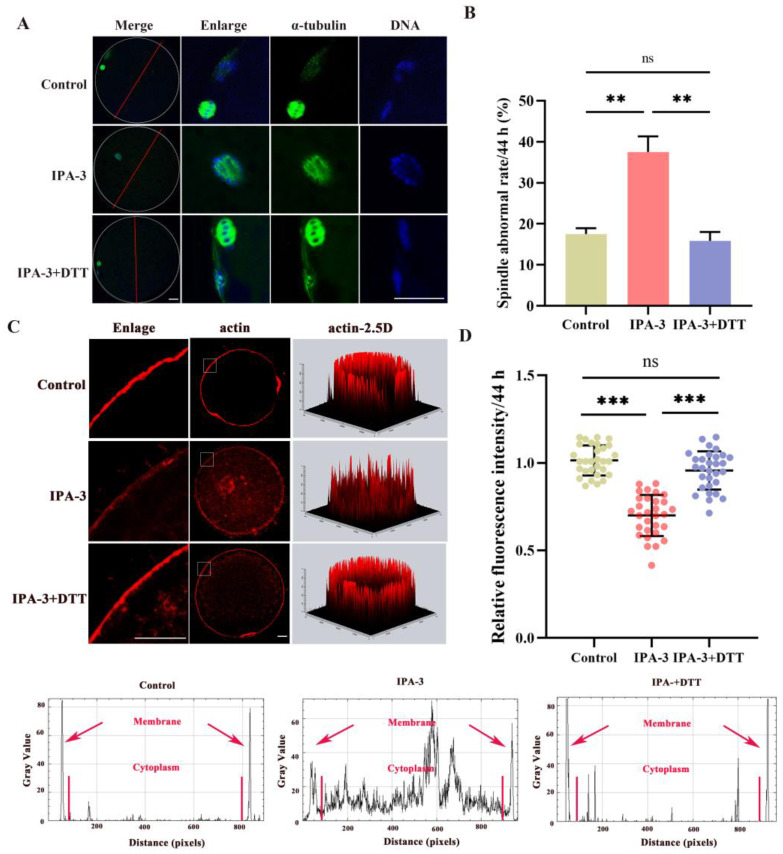
(**A**) Representative images of spindle morphology (green) and chromosome alignment (blue) in different groups after 44 h of culture. Scale bar = 20 μm. (**B**) Percentages of aberrant spindles in different groups. *n* = 120. (**C**,**D**) Representative images and quantification of the immunofluorescence intensity level of actin filaments in different groups. Scale bar = 20 μm. *n* = 30. The letter “*n*” means the total number of oocytes in each group. ns, no significant difference, *p* > 0.05; **, *p* < 0.01; ***, *p* < 0.001.

**Figure 6 biomolecules-14-00237-f006:**
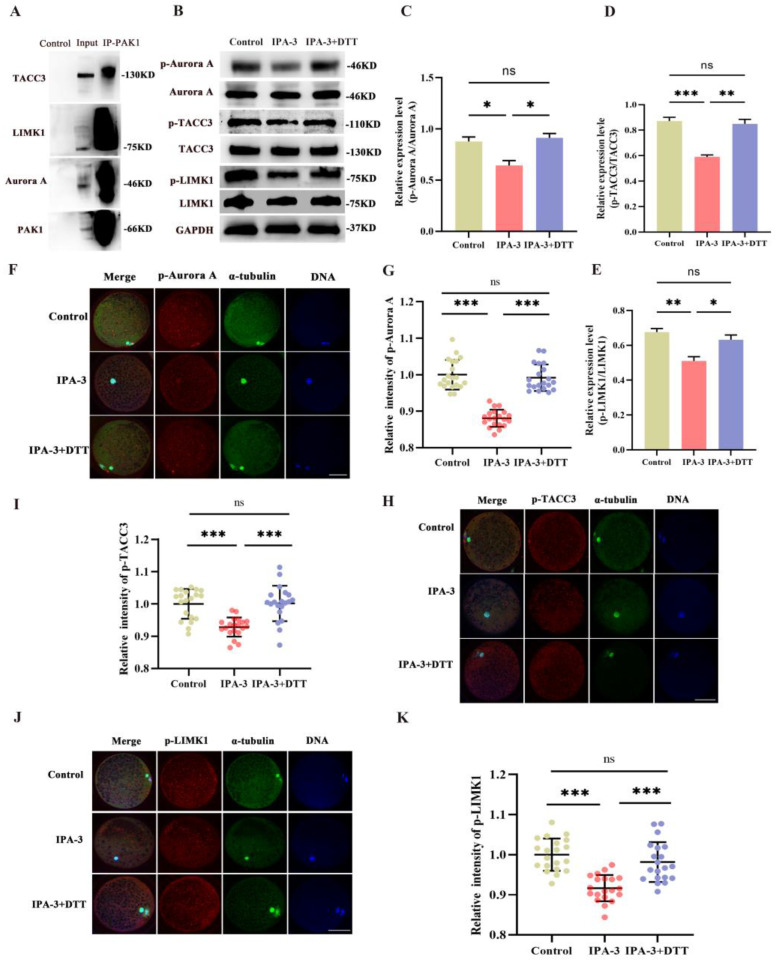
(**A**) Results of co-immunoprecipitation between PAK1 and its substrates, Aurora A, TACC3, and LIMK. (**B**) Representative images of p-Aurora A, p-TACC3, and p-LIMK1 expression in the control, IPA, and IPA-3+DTT groups using immunoblotting. (**C**–**E**) The analysis of p-Aurora A, p-TACC3, and p-LIMK1 expression in different groups. (**F**) Representative images of subcellular localization of p-Aurora A. Red, p-Aurora A; green, α-tubulin; blue, chromosome. Scale bar = 20 μm. (**G**) The quantification of the immunofluorescence intensity of p-Aurora A. *n* = 20. (**H**) Representative images of the subcellular localization of p-TACC3. Red, p-TACC3; green, α-tubulin; blue, chromosome. Scale bar = 20 μm. (**I**) The quantification of the immunofluorescence intensity of p-TACC3. *n* = 20. (**J**) Representative images of subcellular localization of p-LIMK1. Red, p-LIMK1. Green, α-tubulin. Blue, chromosome. Scale bar = 20 μm. (**K**) The quantification of the immunofluorescence intensity of p-LIMK1. *n* = 20. TACC3, transform acidic coiled-coil 3; p-TACC3, phosphorylated transform acidic coiled-coil 3; LIMK1, LIM kinase 1; p-LIMK1, phosphorylated LIM kinase 1. The letter “*n*” means the total number of oocytes examined by immunofluorescence staining in each group. ns, no significant difference, *p* > 0.05; *, *p* < 0.05; **, *p* < 0.01; ***, *p* < 0.001. Original blot images can be found in Appendix A.

## Data Availability

Data will be made available on request.

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
