# Peer review of "PAK1-Dependent Regulation of Microtubule Organization and Spindle Migration Is Essential for the Metaphase I–Metaphase II Transition in Porcine Oocytes"

_biomolecules, 2024, doi:10.3390/biom14020237_

Round 1

Reviewer 1 Report

Comments and Suggestions for Authors

The draft of the paper effectively presents the role of PAK1 in oocyte maturation during the transition from metaphase I (MI) to metaphase II (MII) in porcine oocytes. The study highlights how PAK1, a critical downstream target of small Rho GTPase, is essential in regulating cytoskeletal kinetics, cell proliferation, and migration. It emphasizes PAK1's significance in spindle assembly during the first meiotic division, addressing the unclear role of PAK1 during the MI-MII transition. The research demonstrates that activated PAK1, co-localized with α-tubulin, shows increased expression from MI to MII. However, inhibiting PAK1 with IPA-3 disrupts the extrusion of the first polar body and causes arrest at the anaphase-telophase stage, along with spindle reassembly defects.

The reversal of these adverse effects by DTT and the interactions of PAK1 with Aurora A, TACC3, and LIMK1, further underscore its crucial role in spindle assembly and migration. The paper concludes that PAK1 is vital for accurate spindle assembly and movement during the MI-MII transition, associated with the activities of p-Aurora A, p-TACC3, and p-LIMK1, thereby enriching our understanding of meiosis in porcine oocyte maturation.

Overall, this draft is well-written and adequately addresses the critical role of PAK1 in porcine oocyte maturation. However, a major issue is the lack of a detailed discussion on the specific molecular mechanisms by which PAK1 interacts with Aurora A, TACC3, and LIMK1. This aspect is crucial for a comprehensive understanding of PAK1's role in oocyte maturation and should be elaborated upon in future revisions.

Another problem is as follows:  Figure 1 shows that the maximum decrease in phosphorylated pak1 occurred after 28 hours. However, while the experimental conditions for tracing the subsequent upward trend were performed in several-hour increments, no data were shown for the time period prior to 28 hours, which strongly raises the question, "Could the maximum decrease in phosphorylated pak1 have occurred at an earlier time period? This fact strongly raises the question, "Could it be that the maximum decrease in phosphorylated pak1 occurred earlier in the day? This question has made it difficult for us to verify the experimental data. To clear up this point, it is necessary to clarify the ups and downs of phosphorylated pak1 at several points between 0 and 28 hours, and then mention that the focus of this study was on the period after 28 hours, or else, it is necessary to perform and present hypothesis-testing experiments focusing on earlier time periods as well. presentation is needed.

Comments on the Quality of English Language

There are no special remarks.

Author Response

Ms. Ref. No.: biomolecules-2818843

Title:PAK1-dependent regulation of microtubule organization and spindle migration is essential for the MI-MII transition in porcine oocytes

Dear editors and reviewers,

We appreciate your kind advice and comments concerning our manuscript entitled “PAK1-dependent regulation of microtubule organization and spindle migration is essential for the MI-MII transition in porcine oocytes” (biomolecules-2818843). The advice and comments are all valuable and helpful in revising and improving this paper. The manuscript has been carefully revised according to the comments of the reviewers. All changes to the original manuscript within the document were highlighted using the track changes mode (please see the "revised highlighted" manuscript). The revision and the responses to the reviewer’s comments are addressed point-by-point below.

Responses to Reviewer’s comments

Reviewer #1

Comment 1: Overall, this draft is well-written and adequately addresses the critical role of PAK1 in porcine oocyte maturation.

Response: Thanks for your high recognition of our research.

Comment 2: A major issue is the lack of a detailed discussion on the specific molecular mechanisms by which PAK1 interacts with Aurora A, TACC3, and LIMK1. This aspect is crucial for a comprehensive understanding of PAK1's role in oocyte maturation and should be elaborated upon in future revisions.

Response: Thank you for the valuable comments and helpful suggestion. According to the suggestion, more detailed elaborations of molecular mechanisms by which PAK1 interacts with Aurora A, TACC3, and LIMK1 has been supplemented to the original manuscript, please see line 334-336, 348-351 and 359-361. And new references [34], [40] and [43] was also added in line 457, 567 and 475.

Comment 3: Figure 1 shows that the maximum decrease in phosphorylated pak1 occurred after 28 hours. However, while the experimental conditions for tracing the subsequent upward trend were performed in several-hour increments, no data were shown for the time period prior to 28 hours, which strongly raises the question, "Could the maximum decrease in phosphorylated pak1 have occurred at an earlier time period? This fact strongly raises the question, "Could it be that the maximum decrease in phosphorylated pak1 occurred earlier in the day? This question has made it difficult for us to verify the experimental data. To clear up this point, it is necessary to clarify the ups and downs of phosphorylated pak1 at several points between 0 and 28 hours, and then mention that the focus of this study was on the period after 28 hours, or else, it is necessary to perform and present hypothesis-testing experiments focusing on earlier time periods as well. presentation is needed.

Response: Thanks for your comment. During the metaphase I (MI) to metaphase II (MII) stages of meiosis, the oocytes form bipolar spindles independent of the centrosome, and exhibit an asymmetric division to segregate homologous chromosomes, and then arrest at the MII stage to wait for further fertilization [11]. And any abnormal behaviors in this period can cause chromosomal aneuploidy and embryonic development defects. The purpose of this study is to explore the potential roles of PAK1 and its underlying mechanism during the MI to MII stages in porcine oocytes. Therefore, the focus of this study is on the period of in vitro culture for 28 to 44 hours (28, 36, and 44 hours, when most oocytes were supposed to reach the MI, ATI and MII, respectively [21,22]). The time point on 0 h for culture was also used as a control to assess the dynamic expression and localization of PAK1 in porcine oocytes during MI-MII transition.

In addition, we have carefully checked the English writing and errors in English spelling, grammar, semantics and style of the original manuscript have been carefully corrected. We hope all the revisions will make this paper acceptable.

_______________________________________________________________________________

Once again, we appreciate your carefully and patiently reviewing of our manuscript. And thanks very much for your valuable comments and suggestions. Please feel free to let me know if you have any further questions.

Yours sincerely,

Jan. 27, 2024

_______________________________

Shiqiang Ju, DVM, Ph.D. Professor

College of Veterinary Medicine

Nanjing Agricultural University

1 Weigang, Nanjing, Jiangsu 210095

China

Ph:86-25-8439.5595(o)

Fax:86-25-8439.8669

Email: jusq@njau.edu.cn

Reviewer 2 Report

Comments and Suggestions for Authors

Thank you for your great paper, for this paper, I have a few doubts and comments as follows:

1. My doubt is about how to choose these timing points, such as the experimental design of IPA-3 treatment (22/28/36/44h).

2. The discussion section is too much summary of the article rather than an extended discussion of the project, you may need to revise that.

3. Could you make the images and legends of the same type be saved to the same size (Figure 2.C and G; Figure 6. C/D/G/E)?

Comments on the Quality of English Language

1. line 28 “and transforming acidic coiled-coil 3 (TACC3) to regulate spindle assembly and interact with LIM”, should be "transform".

2. Many sentences are too long and not so clear, so please make them concise and clear, such as "line 40-42", "line 66-68"....

Author Response

Ms. Ref. No.: biomolecules-2818843

Title:PAK1-dependent regulation of microtubule organization and spindle migration is essential for the MI-MII transition in porcine oocytes

Dear editors and reviewers,

We appreciate your kind advice and comments concerning our manuscript entitled “PAK1-dependent regulation of microtubule organization and spindle migration is essential for the MI-MII transition in porcine oocytes” (biomolecules-2818843). The advice and comments are all valuable and helpful in revising and improving this paper. The manuscript has been carefully revised according to the comments of the reviewers. All changes to the original manuscript within the document were highlighted using the track changes mode (please see the "revised highlighted" manuscript). The revision and the responses to the reviewer’s comments are addressed point-by-point below.

Responses to Reviewer’s comments

Reviewer #2

Comment 1: My doubt is about how to choose these timing points, such as the experimental design of IPA-3 treatment (22/28/36/44h).

Response: Thanks for the question. Previous studies have reported that most porcine oocytes develop to metaphase I (MI), anaphase-telophase (ATI) and metaphase II (MII) stages after being cultured in vitro for 28, 36, and 44 hours, respectively [21, 22]. The purpose of this study is to explore the potential roles of PAK1 and its underlying mechanism during the MI to MII stages in porcine oocytes. Therefore, in Figure 2A, the oocytes were treated with IPA-3 at 28 h of culture (MI stage), and the first polar body (PB1) extrusion and the meiotic progression of the oocytes were valued at 44 h of culture (MII stage). From the results, most oocytes were arrested at the ATI stage by IPA-3 treatment at 36 h of culture (ATI stage). So, in the Figure 4A, the oocytes were further treated with DTT to rescue the activity of IPA-3 in porcine oocytes at 36 h of culture (ATI stage).

Comment 2: The discussion section is too much summary of the article rather than an extended discussion of the project, you may need to revise that.

Response: Thanks for the helpful suggestion. According to the suggestion, we have enriched the detailed discussion on the molecular mechanisms by which PAK1 interacts with Aurora A, TACC3, and LIMK1. Please see the section of the discussion.

Comment 3: Could you make the images and legends of the same type be saved to the same size (Figure 2.C and G; Figure 6. C/D/G/E)?

Response: Thanks for your careful check. we have revised the images and legends mentioned above, please see the revised Figure 2.C and G; Figure 6. C/D/G/E.

Comments on the Quality of English Language

Comment 1:line 28 “and transforming acidic coiled-coil 3 (TACC3) to regulate spindle assembly and interact with LIM”, should be "transform".

Response: Thanks for the suggestion. “transforming acidic coiled-coil 3 (TACC3)” has been changed to “transform acidic coiled-coil 3 (TACC3)”, please see line 28.

Comment 2: Many sentences are too long and not so clear, so please make them concise and clear, such as "line 40-42", "line 66-68".

Response: Thanks for the suggestion. According to the reviewer’s suggestion, we revised these sentences mentioned. In addition, we have carefully checked the English writing and errors in English spelling, grammar, semantics and style of the original manuscript have been carefully corrected. We hope all the revisions will make this paper acceptable.

_______________________________________________________________________________

Once again, we appreciate your carefully and patiently reviewing of our manuscript. And thanks very much for your valuable comments and suggestions. Please feel free to let me know if you have any further questions.

Yours sincerely,

Jan. 27, 2024

_______________________________

Shiqiang Ju, DVM, Ph.D. Professor

College of Veterinary Medicine

Nanjing Agricultural University

1 Weigang, Nanjing, Jiangsu 210095

China

Ph:86-25-8439.5595(o)

Fax:86-25-8439.8669

Email: jusq@njau.edu.cn

Round 2

Reviewer 1 Report

Comments and Suggestions for Authors

This manuscript has been fully revised in accordance with the comments. Therefore, this reviewer finds this draft acceptable in its present form.

Author Response

We appreciate your kind recognition of our manuscript.